# Reliability and Efficiency of Metamodel for Numerical Back Analysis of Tunnel Excavation

Yo-Hyun Choi and Sean Seungwon Lee *

Department of Earth Resources and Environmental Engineering, Hanyang University, Seoul 04763, Korea; netisen@hanyang.ac.kr
* Correspondence: seanlee@hanyang.ac.kr; Tel.: +82-2220-2243

**Abstract:** During tunnel construction, the ground properties, initially evaluated, are continuously assessed and verified through back analysis. This procedure generally requires many numerical analyses, so a metamodel based on artificial neural networks has been used to reduce the number of analyses. More datasets can be used to create more reliable metamodels. However, there are no established rules regarding the optimum number of datasets for a reliable metamodel. Metamodels predicting the vertical displacement of the tunnel crown using five ground parameters (unit weight ($\gamma$), uniaxial compressive strength (UCS), material constant $m_i$, geological strength index (GSI), and coefficient of lateral pressure (K)), with 3, 4, 6, 8, and 10 values per property, were created to confirm the reliability of the metamodel based on the number of datasets in this study. Metamodels using 6 and 8 values for each property showed 5% and 1% mean absolute percent errors, respectively. These numbers of each of the properties would be appropriate for developing the metamodel. Among the five parameters, only the results of the global sensitivity analyses of GSI and K are higher than 0.9. According to these results, it is verified that assessments based only on these parameters are sufficient in the back analysis.

**Keywords:** metamodel; artificial neural networks; back analysis; reliability; tunnel excavation

## 1. Introduction

Tunnel engineers use numerical analysis methods, such as the finite element method, finite difference method, and distinct element method to predict the behaviours of underground and structures for support. However, it is difficult to obtain perfect information of ground due to its complexity and uncertainties. Additionally, it is impossible to conduct some experiments to obtain all of the ground properties due to costs and time. Back analysis has been employed to overcome the uncertain and limited information about the ground condition. Back analysis quantitatively assesses the ground properties via numerical analysis using measured displacements [1,2] and stresses [3]. It comprises inverse and direct methods, and the direct method is generally used for the convenience of calculation. In the direct method, numerical analysis is performed, and the displacements or stresses of the analysis are compared with measured displacements or stresses. Here, errors between predicted and measured values are calculated. The numerical analysis is repetitively performed by tuning the target parameters until the mean of the errors is minimised or falls below the target of mean of errors. The object properties of a back analysis can be obtained from the properties that derive results satisfying the tolerance. In this paper, back analysis means the direct method of a back analysis.

Back analysis has been widely used in geotechnical engineering. Gioda and Locatelli [4] conducted a back analysis to assess the elastic modulus and confirmed that the design must consider a lower elastic modulus than the ground investigation results. Fakhimi et al. [5] evaluated the coefficient of lateral pressure and cohesion of the ground



around the tunnel excavation using measure displacements from two small gallery tunnels before the main tunnel construction. Luo et al. [6] performed a three-dimensional back analysis to evaluate the elastic modulus and coefficient of lateral pressure; based on their research, they proposed a change in the length of the tunnel bench to ensure ground stability during construction. However, this method has a disadvantage in that it requires a lot of time because the numerical analyses must be performed repetitively until the allowable range of error is reached. Many researchers have been studying back analysis using artificial neural networks (ANNs) to overcome the above disadvantages. ANN models for back analysis are usually metamodels for numerical analysis. According to Allemang and Hendler [7], a metamodel is defined as "a model to describe another model as an instance". Once a metamodel is created, it has the advantage of reducing the overall back analysis time cause the repetitions of the numerical analyses are not required anymore. Using this method, Yoo and Song [8] assessed the elastic modulus and coefficient of lateral pressure of a tunnel excavation ground, based on 81 numerical analysis results. Song [9] evaluated the elastic modulus, cohesion, and coefficient of lateral pressure using 192 numerical analysis results. In addition, Yoo and Kim [10] increased the number of data and created a metamodel with a total of 2187 numerical analysis results to estimate the ground properties more accurately than that reported by Song [9].

The common point in these previous studies is that the metamodels were generated using two or three values for each of the dependent variables without specific bases. Smaller numbers of values of each of the dependent variables incur less time for the generation of the initial database but cannot guarantee the reliability of a metamodel. On the other hand, the number of values of each dependent variable secures the reliability of a metamodel but additional time is required for the generation of the initial database. Therefore, a study on the number of values required for each dependent variable to guarantee the reliability and efficiency of a metamodel is necessary.

In the present study, the reliability of the metamodel was confirmed according to the number of values of each dependent variable using five ground parameters as the dependent variables and vertical displacement of the tunnel crown as the independent variable. The five ground parameters are as follows: unit weight ($\gamma$); uniaxial compressive strength (UCS); material constant $m_i$; geological strength index (GSI); and coefficient of lateral pressure (K). In addition, the priorities of the parameters used in the metamodel were confirmed through global sensitivity analysis (GSA).

## 2. Methods

This study utilised numerical analyses, ANNs, and GSA. Figure 1 illustrates the flow of this study. First, we generated a database by performing numerical analyses on tunnel excavations. The numerical analyses were performed by changing the values of the five ground parameters 10 times to generate a total of 100,000 datasets. An optimal metamodel was then determined through grid search by changing the number of nodes and activation functions using this database for ANNs.

Next, the numbers of values of the ground parameters were reduced from 10 to 8, 6, 4, and 3. These values allowed creation of datasets of sizes 32,768 ($8^5$), 7776 ($6^5$), 1024 ($4^5$), and 243 ($3^5$). The optimal models for these four groups of datasets were generated like the optimal model for the group with 100,000 datasets. Each optimal model was subsequently tested with 25% of the datasets left for testing when training the ANNs. In addition, the reliability of each metamodel was verified by testing new numerical analysis datasets resulting from the ground parameter values that were not used previously. Finally, the GSA was performed on the 100,000 datasets to confirm the priorities of the five ground parameters for vertical displacement of the tunnel crown.

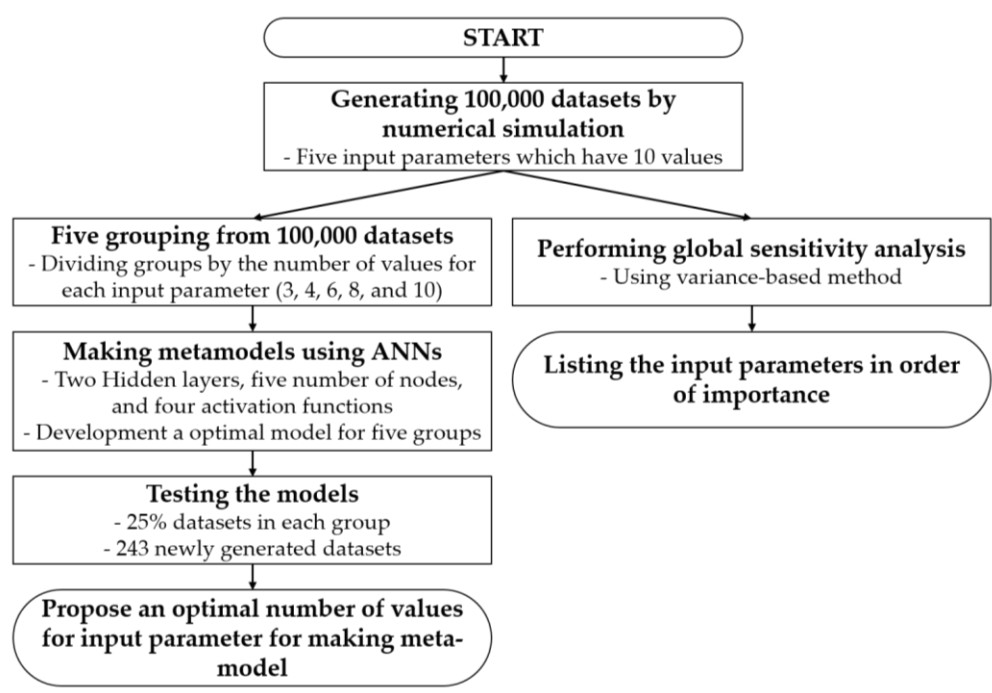

**Figure 1.** Research flow.

## 2.1. Artificial Neural Networks (ANNs)

An ANN is a technique to derive a quantitatively expressed relationship model of cause and effect through a process of minimising the error between the predicted and target values. The model is then used to infer the resulting values from new cause values. This model typically comprises three layers (input, hidden, and output), with nodes in each layer and activation functions used for the nodes in the hidden layer. As shown in Figure 2, the nodes marked with $I$, $H$, and $O$ are connected to nodes in the front and rear layers by weights ($w$) and biases ($b$).

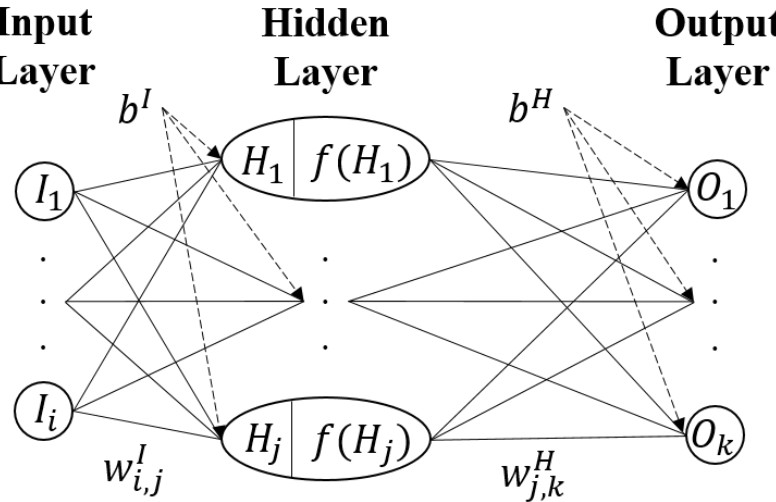

**Figure 2.** General artificial neural network model.

Here, the $I$ nodes allow exporting of the values of the input parameters of the dataset to the hidden layer. Each node $H_j$ of the hidden layer is calculated using Equation (1), and the calculated values are sent to the next layer through the activation function ($f$). Each node $O_k$ of the output layer is then calculated through Equation (2). The weights and

biases are then updated to the partial derivatives of weights and biases for the loss function, respectively until the optimum model with the minimum loss function value is found.

$$H_j = \sum_i I_i \times w^I_{i,j} + b^I \tag{1}$$

$$O_k = \sum_j H_j \times w^H_{j,k} + b^H \tag{2}$$

The numbers of nodes in the input and output layers are determined according to the number of dependent and independent variables, respectively. The number of hidden layers, the number of nodes in each hidden layer, and their activation functions are determined from the form of the optimum model obtained through grid search, which allows the creation of various combinations of models.

### 2.2. Global Sensitivity Analysis (GSA)

GSA is a technique that enables grading-dependent variables that affect an independent variable by considering the changes in the other dependent variables. GSA has a disadvantage in that the analysis method is more complicated than local sensitivity analysis; however, among the GSA approaches, the variance-based (Vb) method based on Sobol's sensitivity index [11] has been used by many researchers owing to its simplicity of interpretation [12]. In the Vb method, sensitivity analysis is performed using Equation (3). Here, $S_{T_i}$ is called total effect and indicates the degree of the overall influence of the dependent variable *i* on the independent variable *Y*. $X_{\sim i}$ is the set of remaining dependent variables except *i* [13]. $S_{T_i}$ is a relative concept, meaning that the larger its value, the stronger its effect on the independent variable.

$$S_{T_i} = 1 - \frac{V[E(Y|X_{\sim i})]}{V(Y)} \tag{3}$$

## 3. Datasets

The Fast Lagrangian Analysis of Continua v7.0 (FLAC), which is a finite-difference-based numerical analysis program developed by Itasca [14] and widely used in tunnel design and research fields, was used to develop a database for the metamodel and for performing the GSA. The Hoek–Brown model was used as the constitutive model, and a total of 100,000 analyses were performed by changing the ten values of the five ground parameters individually, as described later in Section 3.2. The total of 100,000 numerical analyses required 36 days, 18 h, and 37 min for complete execution. The vertical displacements at the tunnel crown according to the 100,000 analyses were in the range of $-1.6 \times 10^{-2}$ to $-3.2 \times 10^{-6}$ m, and the negative sign indicates the direction of gravity.

### 3.1. Ground Parameters

The parameters $\gamma$, UCS, GSI, $m_i$, K, and disturbance factor (D) are required to use the Hoek–Brown model as the constitutive model in FLAC. From these six factors the elastic modulus, Poisson ratio, tensile strength, and constants for the rock mass $m_b$, s, and a, which are required for the numerical analyses, can be calculated [15,16]. Here, D = 0 was used by assuming that all blasting was performed with very good quality control. Finally, the remaining five factors except D were selected as the dependent variables. Similarly to this selection, Lee [17] had selected tunnel depth, UCS, GSI, $m_i$, and K as dependent variables to evaluate the behaviour of rock masses at a deep depth. Except for tunnel depth, four factors are the same as the selected dependent variables in this study. The vertical displacement of the tunnel crown was selected as the independent variable since it is the maximum displacement that occurs around a tunnel and utilised for a back analysis during tunnel construction.

### 3.2. Ranges of the Ground Parameters

Table 1 shows the selected ten values of the five ground parameters; $\gamma$, UCS, and $m_i$ for each rock type investigated by Sharma [18], Palmstrom [19], and Marinos and Hoek [20], respectively. The types of rocks common in these studies were selected as the target rock types in the present study and divided into ten categories, including the maximum and minimum values of each ground parameter. Five igneous rocks (granite, basalt, diabase, granodiorite, and gabbro), two metamorphic rocks (quartzite and gneiss), and three sedimentary rocks (shale, sandstone, and limestone) were selected. GSI values ranging from 0 to 100 [20] were divided into ten classes from 10 to 100. In general, numerical analyses are performed using values of 0.5, 1.0, 1.5, and 2.0 for K. Therefore, it was decided to use ten values from 0.5 to 2.75 for K in this study to include the above four commonly used values. Table 1 shows the values of the ground parameters used in numerical analyses, which $\Delta$ represents the incremental value of each parameter.

**Table 1.** Selected values for the five parameters.

| Number | $\gamma$ [kg/m$^3$] | UCS [MPa] | $m_i$ | GSI | K |
|:---:|:---:|:---:|:---:|:---:|:---:|
| 1 | 2395.5 | 95.0 | 6 | 10 | 0.5 |
| 2 | 2466.9 | 115.6 | 9 | 20 | 0.75 |
| 3 | 2538.2 | 136.1 | 12 | 30 | 1 |
| 4 | 2609.6 | 156.7 | 15 | 40 | 1.25 |
| 5 | 2680.9 | 177.2 | 18 | 50 | 1.5 |
| 6 | 2752.3 | 197.8 | 21 | 60 | 1.75 |
| 7 | 2823.6 | 218.3 | 24 | 70 | 2 |
| 8 | 2895 | 238.9 | 27 | 80 | 2.25 |
| 9 | 2966.4 | 259.4 | 30 | 90 | 2.5 |
| 0 | 3037.7 | 280.0 | 33 | 100 | 2.75 |
| $\Delta$ | 71.3 | 20.6 | 3 | 10 | 0.25 |

### 3.3. Characteristic of the Model

A horseshoe-shaped tunnel was analysed in this study with a 4 m radius and 1.4 tunnel width/height ratio. There are a total of 2735 tunnels in South Korea. The widths of 85% of the total tunnels are distributed from 8 to 14 m according to Road, Bridge and Tunnel Statistics [21]. Additionally, the tunnel width/height ratios of 56% of the total tunnels are distributed from 1.2 to 1.6. It was expected that the 100,000 numerical analyses would require a lot of computational time, so an axisymmetric analysis was performed to shorten the time. Moreover, it was expected that the tunnel radius would greatly influence the total analysis time. Therefore, to confirm our expectations, ten preliminary numerical analyses were performed for tunnel radii of 4, 5, 6, and 7 m. These radii are typically half of the tunnel widths generally distributed in South Korea. The effect of tunnel width/height ratio on the numerical analysis time was not expected to be significant like width. Therefore, the tunnel width/height ratio applied was the median value of 1.4 to generalize the most distributed range of 1.2 to 1.6. The same values of the ground parameters were used for the different tunnel radii, and different values of the ground parameters were used when analysing the same tunnel radius. Based on the ten numerical analyses of the tunnel radii of 4, 5, 6, and 7 m, the performance times were obtained as 5, 13, 24, and 41 min, respectively. These results imply that about 35 days, 90 days, 167 days, and 285 days would be required to perform the 100,000 numerical analyses for the four radii. Thus, the radius was selected as 4 m for efficient analysis since the time difference between tunnel radii of 4 m and 7 m is 250 days; in addition, a difference of 55 days is expected even with the next highest radius of 5 m. The tunnel is located at the lower part of the surface at 40 m, and the lower and right boundaries are set at distances of 36 m each (=4.5 × diameter) from the centre of the tunnel, as shown in Figure 3. The gravity was 9.81 m/s$^2$ in a downward direction, and the X displacements at the left and right boundaries were set to 0, while the Y displacement at the lower boundary was set to 0.

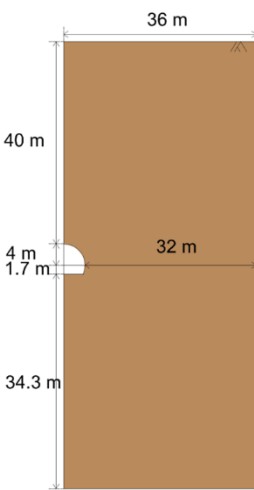

**Figure 3.** Cross section of the numerical analysis.

### 3.4. Grouping Datasets

The 100,000 datasets were grouped into five based on the number of values used per parameter to confirm the reliability of the metamodel. The number of values for each parameter in the five groups were 3, 4, 6, 8, and 10, which were defined as groups A, B, C, D, and E, respectively. For the consistency of the comparison in this study, an identical range of five ground parameters for the five dataset groups was set. Additionally, the numbers of input value were reduced by two in each step so that the values of each parameter can be evenly distributed the within the range. In the last step, the authors chose three values per parameter rather than two, because we believed the use of two input values would increase the error over three input values. Table 2 shows the numbers of values used for each group, as described in Table 1. Here, O means usage and - means exclusion.

**Table 2.** Numbers of the values used in each group.

| Number of Value | 1 | 2 | 3 | 4 | 5 | 6 | 7 | 8 | 9 | 10 |
|---|---|---|---|---|---|---|---|---|---|---|
| Group A | O | - | - | - | O | - | - | - | - | O |
| Group B | O | - | - | O | - | - | O | - | - | O |
| Group C | O | - | O | - | O | O | - | O | - | O |
| Group D | O | O | O | - | O | O | - | O | O | O |
| Group E | O | O | O | O | O | O | O | O | O | O |

## 4. ANNs-Based Metamodel

The five ground parameters, namely $\gamma$, UCS, GSI, $m_i$, and K, were used as the input parameters, and the vertical displacement of the tunnel crown was used as the output parameter to develop the metamodels using ANNs. Jung et al. [22] noted that it was appropriate to use two hidden layers for ANNs; accordingly, two hidden layers were used in this study. The optimum model was determined by grid search for the number of nodes and activation functions used in each hidden layer. For the grid search, the numbers of nodes were determined using the five equations shown in Table 3, where $N_i$ is the number of input parameters. The number of nodes used for the grid search was 8 (rounding off at 7.5), 11, 15, 20, and 25. As activation functions, sigmoid, hyperbolic tangent, ReLU, and Leakyrelu functions were for the grid search and $\alpha = 0.2$ was used for Leakyrelu. The training, verification, and test datasets were generated by dividing each dataset into 60%, 15%, and 25% of the total for each group. The early stopping condition was a situation in which the minimum value of the loss function was not updated for 500 iterations. Root mean square error was used as the loss function, as expressed in Equation (4). Here, $\delta_{pi}$ and $\delta_{ti}$ are the predicted and target vertical displacement of the tunnel crown. A total of 400 models ($5 \times 5 \times 4 \times 4$) were generated for each group through grid search, and the

model with the lowest derived value of the loss function among the 400 models was selected as the optimal model for the group. This optimal model for each group is a metamodel representing FLAC.

$$\text{Loss} = \sqrt{\frac{1}{n}\sum_{i=1}^{n}\delta_{ti}-\delta_{pi}} \tag{4}$$

**Table 3.** Equations for determining the numbers of nodes.

| Equation | Reference |
|---|---|
| $3N_i/2$ | Mamaqani [23] |
| $2N_i + 1$ | Nielsen [24] |
| $3N_i$ | Hush [25] |
| $4N_i, 5N_i$ | Choi and Lee [26] |

## 5. Results and Discussions

### 5.1. Metamodel and Prediction Results for Test Datasets

The structure of the metamodel of each group is expressed in Table 4. Metamodels developed from A to E groups are named Metamodels A, B, C, D, and E, respectively. Loss values were expected to decrease as the number of values increased but the loss value of Metamodel A was lower than those of Metamodels B and C. This situation can be considered a type of noise called overfitting. Overfitting means that the ANNs model has not been generalized since it is so accurately tailored to the training and verification datasets. Figure 4 shows the minimum, maximum, and average losses and mean absolute percent error (MAPE) of a metamodel of each group. In Figure 4, min and max imply the lowest and highest loss values in 400 models developed for grid search of each group. Average implies the average loss value of the 400 models of each group. These three indexes are based on the left Y-axis. MAPE is the evaluation index expressed by Equation (5). The MAPE represented by yellow diamonds is based on the right Y-axis.

$$\text{MAPE} = \frac{1}{n}\sum_{i=1}^{n}\left|\frac{\delta_{ti}-\delta_{pi}}{\delta_{ti}}\right| \tag{5}$$

**Table 4.** Structures of the metamodels and test results.

| Metamodel | Number of Nodes | | Activation Function | | Loss (m) | MAPE (%) |
|---|---|---|---|---|---|---|
| | 1st Hidden Layer | 2nd Hidden Layer | 1st Hidden Layer | 2nd Hidden Layer | | |
| A | 20 | 8 | S | R | 0.007341 | 1.46 |
| B | 8 | 20 | L | R | 0.01434 | 1.84 |
| C | 8 | 11 | R | H | 0.01395 | 1.27 |
| D | 20 | 20 | H | S | 0.00552 | 0.50 |
| E | 25 | 20 | H | R | 0.00329 | 0.37 |

H: hyperbolic tangent; L: Leakyrelu; R: ReLU; S: sigmoid.

Except for the minimum value, as we expected, the loss of the average and maximum value decreased as the number of values per property increased. These tendencies imply that Metamodel A is a special case of predicting validation datasets well. The prediction degree of the metamodels was evaluated using test datasets of each group by MAPE. Every metamodel shows a MAPE value of less than 2%. This result could imply that the metamodel created using three values per parameter is also reliable for representing FLAC. However, all of the parameter values in the test datasets exist in the training datasets. It is possible that these metamodels are good at predicting only for datasets with property values existing in training datasets. That is, overfitting must be checked. Therefore, further verifications are required using new datasets that utilize other property values.

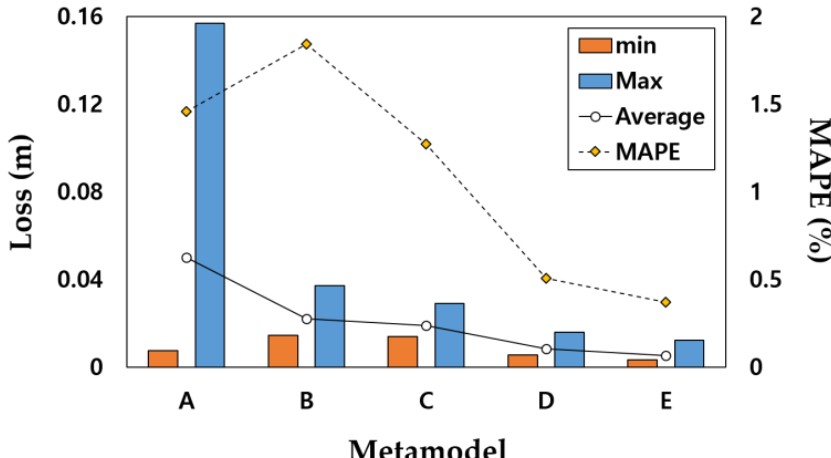

**Figure 4.** Results of loss and MAPE of each metamodel.

*5.2. Verification of the Metamodels*

To verify the metamodels, 243 datasets with parameter values that were not used in model training were generated by FLAC. Value numbers 11, 12, and 13 in Table 5 were used in the new test dataset, and these are the mean values of the numbers 1 and 2, 5 and 6, and 9 and 10 in Table 1, respectively. The numerical analysis method is the same as that noted in Section 3.

**Table 5.** Additional values of the five parameters to generate new test datasets.

| Number | $\gamma$ [kg/m³] | UCS [MPa] | $m_i$ | GSI | K |
|---|---|---|---|---|---|
| 11 | 2431.2 | 105.3 | 7.5 | 15 | 0.625 |
| 12 | 2716.6 | 187.5 | 19.5 | 55 | 1.625 |
| 13 | 3002.1 | 269.7 | 31.5 | 95 | 2.625 |
| $\Delta$ | 285.5 | 28.2 | 11.5 | 40 | 1 |

The verification results are shown in Figure 5, and the MAPEs of Metamodel A to E are 9.5%, 6.1%, 4.9%, 0.8%, and 1.2%, respectively. The MAPE of Metamodel A shows the highest error (near 10%) among those of the five groups. From this result, it is assured that Metamodel A has an overfitting problem. That is, a metamodel developed by three values per parameter is unsuitable for computing datasets with new parameter values, which are not used in the training model, instead of the FLAC. The MAPE tends to decrease as the number of values per parameter increases and from Metamodel C, which was developed by six values per parameter, MAPE falls below 5%. Similar to predicting test datasets, Metamodels D and E developed by eight and ten values per parameter show very low MAPE of about 1%. These results imply that the generalization and performance of Metamodels D and E are excellent since they have been trained with a number of datasets. This is attributed to the MAPE converging to 1% as the number of values of each parameter increases.

The MAPE trends in Figures 4 and 5 show a similar trend of decreasing as the number of values per parameter increases. In the trend, there are two different points. The ranks of the magnitude of the MAPE between Metamodels A and B and between D and E are reversed. Metamodel A shows a slightly lower error rate than B when predicting the test dataset in Section 5.1 due to the overfitting of Metamodel A. Metamodels D and E show low MAPE in both tests since they have a good generalization and good prediction performance as mentioned above. Therefore, the MAPE results are converged for the two tests for Groups D and E.

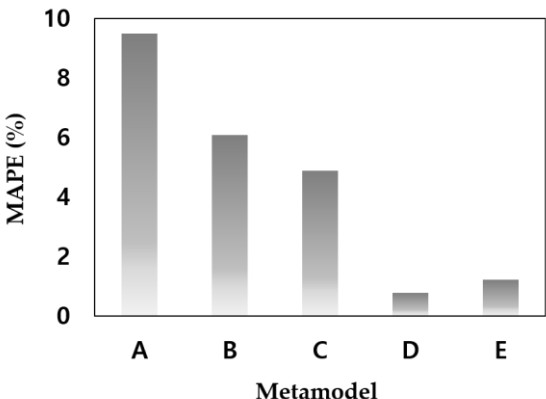

**Figure 5.** MAPE of each metamodel for the new test dataset.

Based on the computing time to make the database of a metamodel of Group E, the computing times for Group A to D are expected about 0.2%, 1%, 8%, and 33%, respectively. These expectations considered only total numerical analyses time for database without metamodel development time using ANNs. The more datasets, the more time it takes to develop a model. If we allow about 5% or 1% error of a metamodel, then we can develop a metamodel in 8% or 33% of the time required to develop a metamodel using 10 values per parameter. These results will be helpful in efficiently generating a metamodel.

### 5.3. Global Sensitivity Analysis (GSA)

The GSA was performed using the Vb method to analyse the effects of the five ground parameters on the vertical displacement of the tunnel crown. The 100,000 datasets of Group E were used, and the results are shown in Figure 6. The $S_{T_i}$ of $\gamma$, UCS, $m_i$, GSI, and K are 0.015, 0.024, 0.045, 0.929, and 0.995, respectively. Among them, $S_{T_i}$ of GSI and K show extremely high values (exceeding 0.9) that exceed the average of $S_{T_i}$ of other parameters. This result means that it is enough to consider only GSI and K among the five parameters when tunnel engineers conduct a back analysis. In the numerical analyses in this study, the elastic modulus value was calculated from the GSI. Therefore, this result is consistent with the selection of the elastic modulus and lateral pressure coefficient as the parameters of the back analyses, considering the vital parameters of tunnel displacements in existing studies [4–6].

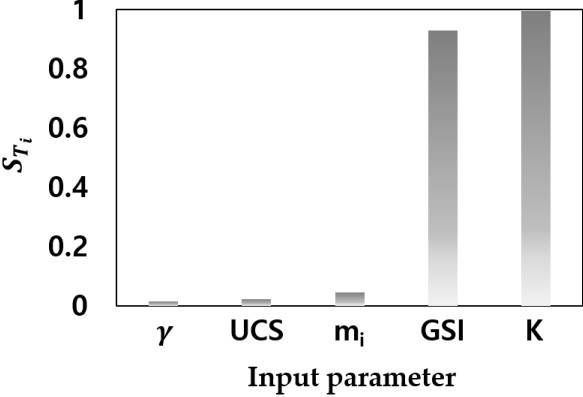

**Figure 6.** Results of global sensitivity analysis for five input parameters.

### 6. Conclusions

In this study, the reliability of an ANN-based metamodel used for tunnel back analysis was analysed according to the number of values of each ground parameter. The real values of the ground parameters are not constant owing to their inherent uncertainties. Considering this, it is recommended to use a minimum of six values per parameter showing

results in a MAPE of 5% since it takes only 8% of the numerical analyses time using 10 values per parameter. Furthermore, it is recommended to use eight values per parameter to obtain higher reliability of the metamodel, although this requires four times as much computational time as using six values. Even so, this time is only 33% of the time required for a metamodel developed by 10 values per parameter. From the results of the GSA, it was confirmed that the $S_{T_i}$ of the $\gamma$, UCS, and $m_i$ are lower than 0.05, otherwise the $S_{T_i}$ of the GSI and K exceed 0.9. It implies that GSI and K must be included in a back analysis as dependent parameters. These results show the priorities of only the ground parameters around a small radius tunnel of 4 m, so additional sensitivity analyses, including support parameters, are required. Therefore, a sensitivity analysis study is being conducted by considering GSI, K, and variables of the supports as dependent variables for a tunnel with a radius of 7 m using six values per parameter based on the results of the present study. In addition, we will attempt to create a metamodel of a tunnel construction site using six values and verify the applicability of this result.

This study was undertaken to find the required minimum number of values per dependent parameter that can secure the reliability of the metamodel. According to the results of this research, geotechnical engineers can efficiently develop a metamodel using six or eight values for each dependent parameter during the design phase. This could reduce time by 92% and 67%, respectively, from the time required when using ten values. Additionally, these metamodels will immediately derive reliable results with a range of 5% and 1% error rates from the numerical analysis results. Therefore, in the tunnel construction stage, the values of ground parameters in abnormal situations can be quickly and reliably evaluated through inverse analysis using the developed metamodel. In this regard, it is possible to quickly apply initial countermeasures, such as changing the support pattern to secure stability.

**Author Contributions:** Data curation, Y.-H.C.; Formal analysis, Y.-H.C.; Investigation, Y.-H.C.; Methodology, Y.-H.C.; Project administration, Y.-H.C.; Validation, Y.-H.C.; Writing—original draft preparation, Y.-H.C.; Conceptualization, S.S.L.; Funding acquisition, S.S.L.; Project administration, S.S.L.; Supervision, S.S.L.; Writing—review and editing, S.S.L. All authors have read and agreed to the published version of the manuscript.

**Funding:** This work was supported by the Korea Agency for Infrastructure Technology Advancement (KAIA) grant, funded by the Ministry of Land, Infrastructure and Transport (22UUTI-C157786-03) and by the National Research Foundation of Korea (NRF) grant, funded by the Korean government (MSIT) (NRF-2020R1A6A3A13077513).

**Institutional Review Board Statement:** Not applicable.

**Informed Consent Statement:** Not applicable.

**Data Availability Statement:** Not applicable.

**Conflicts of Interest:** The authors declare no conflict of interest. The funders had no role in the design of the study; the collection, analyses, or interpretation of data; the writing of the manuscript; or the decision to publish the results.

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
