# Peer review of "Reliability and Efficiency of Metamodel for Numerical Back Analysis of Tunnel Excavation"

_applsci, doi:10.3390/app12146851_

Round 1

Reviewer 1 Report

In this paper, the reliability of the metamodel is confirmed according to the number of values of each dependent variable, using five soil parameters as dependent variables and the vertical displacement of the tunnel crown as the independent variable.

The following problems exist in the paper:

The first part of the article is insufficient - the state of the art and list of references too short.

There should be an explanation of the back method.

Figures 1 and 2 could be better arranged in the article and figure 4 is unnecessary. 

Author Response

Thanks

Sean

Reviewer 2 Report

1. One of the biggest concern of mine is about the application.

How can this methodology support practitioners? Which can be the application? How does your research may support engineers during the design phase?

2. Sec.2 l.75-77

On which bases the authors reduced the number of paraments?

3. Sec. 3 l.129-130

In the authors opinion, is the magnitude of these displacements acceptable?

4. Sec. 3.1 l.138-139

What is the reason behind the choice of this independent  parameter?

5. Sec. 3.3 l.158-161

This sentence is very confusing. Please rephrase it.

6. Sec. 3.3 l.168

Why is the median value of 1.4 chosen?

7. Sec. 3.3 l.174

Have the authors any idea/suggestion to reduce this computational time of 250 days? It would be useful to have in the paper a small paragraph on way to address the reduction of computational time and cost.

8. Sec.5

Why presenting only results for 25% Test Datasets?

A comparison between this and another percentage would be useful, especially to strengthen your discussion section.

9. Figure 5

This figure needs to be well explained in the text. It is very confusing the interpretation of it.

10. Sec. 5.2 l.250-252

The conclusion on the group A needs to be justified and explained.

Same for group D and E (l.252-256)

This explanation is not sufficient.

11. Figure 6. 

The MAPE of Figure 6 is not in line with the trend in Figure 5. Please iterate.

12. Sec.6

Your ANN metamodel suggest 3 values per parameters, while you suggest a minimum of 6 parameters due to the uncertainties of the in-situ situation. On which bases you do this recommendations?

13. Sec. 6

Conclusions needs to include the take home message of some of the unclear points mentioned above.

Author Response

Thanks

Sean

Reviewer 3 Report

(1)         In 5.3 section,whether can different K values all express the same importance with the expression  “meaning that each parameter has a significant effect on the vertical displacement of the tunnel crown”? The author is suggested to give a more reasonable and convincing explanation. In addition, whether the accuracy of value of K needs to be discussed?

(2)         Is there any methodological innovation in the research work of the thesis? Are the reference methods applied directly or improved?

(3)         Are Artificial Neural Networks (ANNs) and Global Sensitivity Analysis (GSA) necessary references required? Please confirm.

(4)         Is there any measured data that can enhance the demonstration?

(5)         Whether the research method or parameter selection method can be compared with the existing research to further illustrate the accuracy of the conclusion.

Author Response

Thanks

Sean

Round 2

Reviewer 2 Report

The authors have addressed all my concerns.

I do not have any further comments.